# Psychosocial impacts of COVID-19 in the Guinean population. An online cross-sectional survey

Almamy Amara Touré[1,2]*, Lansana Mady Camara[2], Aboubacar Sidiki Magassouba[3], Abdoulaye Doumbouya[1], Gnoume Camara[3‡], Alsény Yarie Camara[1], Gaspard Loua[4‡], Diao Cissé[3‡], Mohamed Sylla[1], Alpha Oumar Bereté[1‡], Abdoul Habib Beavogui[1,5]

1 Centre National de Formation et de Recherche en Santé Rurale de Maferinyah, Forecariah, Guinea, 2 Université Koffi Annan de Guinée, Conakry, Guinée, 3 Department of Public Health, Faculty of Health Sciences and Techniques, Gamal Abdel Nasser University, Conakry, Guinea, 4 Johns Hopkins Program for International Education in Gynaecology and Obstetrics (JHPIEGHO-Guinée), Conakry, Guinea, 5 Département de Parasitologie–Mycologie, Faculté des Sciences et Techniques de la Santé, Université Gamal Abdel Nasser de Conakry, Conakry, Guinée

☊ These authors contributed equally to this work.
‡ These authors also contributed equally to this work.
* almamy@maferinyah.org

**Data Availability Statement:** All relevant data are within the manuscript and Supporting Information files.

## Abstract

Guinea, like many other African countries, has been facing an unprecedented COVID-19 outbreak, since March 2020. In April 2020, Guinean National agency for health security recorded 1351 confirmed cases of COVID-19, including 313 recoveries and 07 deaths. To address this health crisis, some drastic measures were implemented to prevent the spread of COVID-19. Those measures might potentially cause some psychological problems among Guineans. Thus, we conducted this study to assess the psychosocial impacts of COVID-19 in the Guinean population. We carried out an online cross-sectional survey among internet users in Guinea. A free e-survey platform was used, and questionnaires were sent to internet users. The study ran from May 1 through May 10 2020. Participation in the study was voluntary. Data collection was based on sociodemographic information and self-reported questionnaires: Impact of Event Scale-Revised (IES-R) for stress evaluation, Penn state worry questionnaire (PSWQ), and an adapted Social Psychological Measurements of COVID-19. A total of 280 participants took part in the study; responses from 5 participants were deleted because of incompleteness. The average age of participants was 28.9 [95% CI: 28.1;29.6]. Most of participants were male 65.5% [95% CI: 59.5%;71.1%]. Unemployed participants stood for 48.7% [95% CI: 42.7%;54.8%]. IES-R scale for stress evaluation yielded the following findings: 19.6% (mild), 5.23% (moderate) and 9.15% (severe); 82.8% and 17.2% of participants had respectively reported low and moderate worry. No significant statistical association was found between sociodemographic variables and traumatic events (IES-R and PSWQ). However, 82% of our participants had to cope with the negative impacts of COVID-19. Although there were few cases of traumatic events, negative impacts of COVID-19 on study participants deserve to be underlined. So, further investigations are necessary to identify and disentangle specific psychosocial problems in different Guinean socio-cultural contexts.

**Funding:** This study was self-funded and also supported in the form of administrative support from Maferinyah training and research centre in rural health and the Koffi Annan University of Guinea.

**Competing interests:** The authors have declared that no competing interests exist.

## Introduction

Coronavirus disease 2019 (known as "COVID-19") started in December 2019 in China and has been spreading worldwide with its unparalleled consequences in the recent history of humanity. In the wake of this overwhelming situation, the World Health Organization (WHO) declared the outbreak of COVID-19 as a Public Health Emergency of International Concern and could be characterized as a pandemic [1]. Since then, the number of cases and deaths due to COVID-19 has been ever-increasing, thereby threatening global health. According to the WHO report, globally, 7 410 510 cases of coronavirus has been confirmed with 418 294 deaths toll. African countries have reported 155 762 cases with 3 700 deaths toll [2]. Guinea, like other African countries, has been facing COVID-19 outbreak since March 2020. As of April 28, Guinean National Agency for Health Security recorded 1351 confirmed cases of COVID-19, including 313 recoveries and 07 deaths [3].

The first measures taken by the Guinean authorities were as follows: declaration of the state of health emergency, closure of schools and places of worship (mosques and churches), reduction in the number of passengers in public transportation, ban on the gathering of more than 20 people, obligation to wear masks in public places, and the respect of social distancing. In addition to these stringent measures, Conakry Capital city (deemed as outbreak hotspot) was isolated from the other cities with limited access. Those measures have been reinforced by massive sensitization, the sharing of sanitary kits, and the establishment of a curfew. Moreover, to mitigate the socio-economic impact of coronavirus, the Guinean Government announced a response plan against the pandemic by focusing on support for health and social management.

The aforementioned restrictive measures could potentially have some impacts on the Guinean population, for this is their first time facing such a colossal health crisis. For instance, religious communities could never imagine the closure of places of worship, for, during the 2014 Ebola virus disease outbreak, they never experienced these kinds of restrictive measures. Therefore, this may arguably lead to social and psychological crisis. Since drastic regulations were implemented, it could stem multiple responses from the population according to their features. One of the common reactions to the pandemic is the fear of its psychological impacts [4–7]. Psychological impacts may vary according to the sociodemographic characteristics. A study done in China found that COVID-19 Peritraumatic Distress Index (CPDI) had been associated with gender, age, education, occupation and region [8]. WHO has also mentioned that one of the possible outcomes of COVID-19 pandemic is generating stress and anxiety among people [1].

Furthermore, many studies emphasized the psychological concerns of COVID-19 among the population [9–18]. Management of psychological effects is essential to prevent unexpected events and mitigate COVID-19 effects among people with health underline conditions and those at high risks of being contaminated by COVID-19. Therefore, the primary objective of this study was to assess the psycho-traumatic impact of COVID-19 in the Guinean population. Specifically, we aimed to:

✓ Measure the prevalence of traumatic events (stress, worry) among the Guinean population,

✓ Spot the sociodemographic factors associated with the trauma caused by COVID 19,

✓ Appreciate population perception regarding the measures implemented by the Government in the battle against COVID 19.

## Methodology

We carried out an online cross-sectional survey among literate people (from secondary school to university l). A free online e-survey (https://esurv.org) platform was used to send questionnaires to internet users, and they were encouraged to pass it to one other through social media (Facebook, WhatsApp, messenger) by using snowball sampling. The study population was diverse, and participants came from all sectors of activity in Guinea. However, active internet users were predominant, for our period of study was short. It ran from May 1 to May 10 2020. Participation in the study was voluntary. Participants should have been at least 18 years of age. Institutional approval was obtained from the scientific medical committee of Koffi Anna University in Conakry/Guinea (025/UKAG/P8/2020).

### Study tools

Sociodemographic information: Age, gender, marital status, Current residence, job or occupation, levels of education, source (s) of information (s) about COVID-19, time spent at home to avoid contamination.

Penn State Worry Questionnaire (PSWQ): it is a self-administered, 16-items using Likert-type scale designed to measure worry, Possible ranges of the score are 16–80 with the algorithm of total scores: 16–39 = Low Worry, 40–59 = Moderate Worry, and 60–80 = High Worry [19].

Impact of Event Scale-Revised (IES-R): self-report questionnaire, with 22 questions. The score is interpreted as follow: 24–32: Posttraumatic stress disorder (PTSD) as a clinical concern.33-38: This is considered as the best threshold for a probable diagnosis of PTSD; 39 and above as a Severe PTSD [20].

Adapted Social Psychological Measurements of COVID-19: financial impact (difficulties in meeting basic needs such as foods, means of transportation, and healthcare; loss of a job, no financial impact); support (Giving money to support COVID-19 crisis, supporting government initiatives, supporting the idea of more researches on COVID-19 in Guinea, supporting Government restrictive measures, and need of a strong administration); punishment (penalty for non-respect of preventive measure, including wearing masks, social distancing, ban on gathering of more than 20 people, and the curfew) [21].

### Survey description

A total of 84 parameters were recorded, and it took about 20 minutes to complete them. Each webpage showed 4–6 questions. Each participant could see a total number of pages equal to 5. Participants were allowed to start the survey and finish it at any time.

### Data management and statistical analysis

Duplicated data were carefully handled by blocking the same IP responses. Raw data were extracted in excel format after study completion. Descriptive, statistical analyses were performed for the sociodemographic and all other variables (worry, IES-R). Confidence interval (95%) was built for all variables. We used the chi-square test to identify sociodemographic factors associated with trauma caused by COVID 19. The Likert analysis was used to appreciate population opinions related to the measures implemented by the Guinean government in the fight against COVID 19. We performed all analyses by using R software version 3.6.2. Statistical tests were considered significant when $p < 0.05$.

## Results

### Description of the sample study

A total of 280 participants took part in this study. Data from 5 five participants were deleted because of a great amount of missing data. The average age of our participants was 28.9 [95% CI: 28.1;29.6]; most of them were male, 65.5% [95% CI: 59.5%;71.1%]; single participants were the most represented. Most of our participants lived in Ratoma district. Unemployed participants stood for 48.7% [95% CI: 42.7%;54.8%]. Most of participants had university level 95.3% [95% CI: 92.1%;97.5%]. Participants having spent more time at home per day due to COVID-19 accounted for 86.2% [95% CI: 81.5%;90.0%] (Table 1).

### Prevalence of psychological events

Table 2 shows prevalence of psychological events. More than half of the participants 153 (55.63%%) filled out the Impact of Event Scale–Revised (IES-R) questions and 186 (67.63%) of

**Table 1. Study sample description.**

| Variables | Frequency | % [95% CI] | N |
|---|---|---|---|
| **Age** | | | 275 |
| [18,39] | 260 | 94.5% [91.2%;96.9%] | |
| [40,62] | 15 | 5.45% [3.08%;8.84%] | |
| **Gender** | | | |
| Male | 180 | 65.5% [59.5%;71.1%] | |
| Female | 95 | 34.5% [28.9%;40.5%] | |
| **Marital Status** | | | 275 |
| Married | 92 | 33.5% [27.9%;39.4%] | |
| Single | 183 | 66.5% [60.6%;72.1%] | |
| **Residence** | | | 275 |
| Dixinn | 15 | 5.45% [3.08%;8.84%] | |
| Kaloum | 1 | 0.36% [0.01%;2.01%] | |
| Matam | 3 | 1.09% [0.23%;3.15%] | |
| Matoto | 72 | 26.2% [21.1%;31.8%] | |
| Others* | 35 | 12.7% [9.03%;17.3%] | |
| Ratoma | 149 | 54.2% [48.1%;60.2%] | |
| **Occupation** | | | 275 |
| Civil servant | 39 | 14.2% [10.3%;18.9%] | |
| Freelance | 57 | 20.7% [16.1%;26.0%] | |
| Housewife | 2 | 0.73% [0.09%;2.60%] | |
| Private employee | 43 | 15.6% [11.6%;20.5%] | |
| Unemployed | 134 | 48.7% [42.7%;54.8%] | |
| **Education** | | | 275 |
| High school | 5 | 1.82% [0.59%;4.19%] | |
| College | 8 | 2.91% [1.26%;5.65%] | |
| University | 262 | 95.3% [92.1%;97.5%] | |
| **Time spent at home (hours/day)** | | | 275 |
| ≤ 8 | 38 | 13.8% [9.97%;18.5%] | |
| > 8 | 237 | 86.2% [81.5%;90.0%] | |
| **Legend** | | | |
| **N = Total Frequencies** | | | |

*Participants from outside Conakry capital city.

**Table 2. Psychological impact of COVID-19 pandemic.**

| Variables | Frequency | % [CI 95%] | N |
|---|---|---|---|
| IES-R | | | 153 |
| Normal | 101 | 66.0% [57.9%;73.5%] | |
| Mild | 30 | 19.6% [13.6%;26.8%] | |
| Moderate | 9 | 5.23% [2.28%;10.0%] | |
| Severe | 13 | 9.15% [5.09%;14.9%] | |
| PSWQ | | | 186 |
| Low worry | 154 | 82.8% [76.6%;87.9%] | |
| Moderate worry | 32 | 17.2% [12.1%;23.4%] | |

the participants completed the Penn State Worry Questionnaire (PSWQ). IES-R yielded for the stress evaluation the following findings: mild (19.6%) [95% CI: 13.6%;26.8%], moderate (5.23%) [95% CI: 2.28%;10.0%] and severe (9.15%) [95% CI: 5.09%;14.9%] (severe); while 82.8% [95% CI: 76.6%;87.9%] and 17.2% [95% CI: 12.1%;23.4%] of participants respectively reported low and moderate worry (PSWQ).

## Participants' sources of information about COVID-19

Participants were asked to select three sources of information; most of the respondents chose as follows: Facebook, World Health Organization (WHO) website, and other private websites (Fig 1).

## Psychological associated factors

No statistical association was found between sociodemographic variables and traumatic events (IES-R and PSWQ) (Tables 3 and 4).

## Population perception regarding the government measures in the fight against COVID-19

About 82% of our participants had to cope with adverse impacts of COVID-19, and 64% of participants spent a hard time getting means of transportation, we found that 54% of participants had lost their job, and only 12% had been spared from the financial impact of COVID-19 (Fig 2). Almost all the participants responded that they wanted more researches on COVID-19 in Guinea (94%); the majority of participants (90%) supported both Government

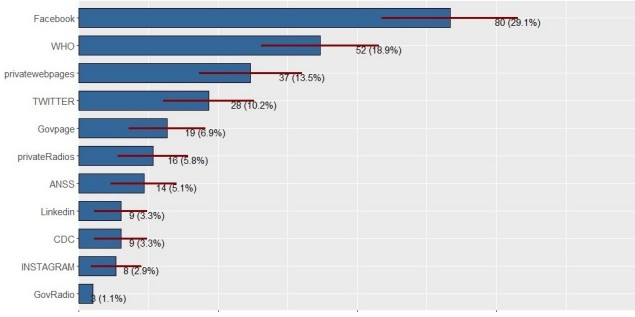

**Fig 1. Source of COVID-19 information.** Govpage = Government web page; ANSS = abbreviation in French which means "National Agency for Health Security"; CDC = Centers for Disease Control and Prevention (United States); GovRadio = Government Radio station.

**Table 3. Bivariate analysis Impact of Event Scale-Revised (IES-R) and sociodemographic characteristics.**

|  | Normal | Mild | Moderate | Severe | p-value |
|---|---|---|---|---|---|
|  | N = 101 | N = 30 | N = 8 | N = 14 |  |
| Age |  |  |  |  | 0.137 |
| 18–39 | 97 (96.0%) | 27 (90.0%) | 8 (88.9%) | 11 (84.6%) |  |
| 40–62 | 4 (3.96%) | 3 (10.0%) | 1 (11.1%) | 2 (15.4%) |  |
| **Gender** |  |  |  |  | **0.578** |
| Male | 63 (62.4%) | 19 (63.3%) | 3 (37.5%) | 8 (57.1%) |  |
| Female | 38 (37.6%) | 11 (36.7%) | 5 (62.5%) | 6 (42.9%) |  |
| **Marital Status** |  |  |  |  | **0.435** |
| Married | 36 (35.6%) | 13 (43.3%) | 1 (12.5%) | 6 (42.9%) |  |
| Single | 65 (64.4%) | 17 (56.7%) | 7 (87.5%) | 8 (57.1%) |  |
| **Residence** |  |  |  |  |  |
| Dixinn | 4 (3.96%) | 3 (10.0%) | 0 (0.00%) | 1 (7.14%) |  |
| Matoto | 29 (28.7%) | 8 (26.7%) | 6 (75.0%) | 4 (28.6%) |  |
| Outside Conakry | 13 (12.9%) | 5 (16.7%) | 0 (0.00%) | 2 (14.3%) |  |
| Ratoma | 55 (54.5%) | 14 (46.7%) | 2 (25.0%) | 7 (50.0%) |  |
| **Occupation** |  |  |  |  |  |
| Civil servant | 15 (14.9%) | 2 (6.67%) | 1 (12.5%) | 4 (28.6%) |  |
| Freelance | 17 (16.8%) | 5 (16.7%) | 0 (0.00%) | 4 (28.6%) |  |
| Housewife | 0 (0.00%) | 0 (0.00%) | 0 (0.00%) | 1 (7.14%) |  |
| Private sector employee | 20 (19.8%) | 7 (23.3%) | 3 (37.5%) | 1 (7.14%) |  |
| Unemployed | 49 (48.5%) | 16 (53.3%) | 4 (50.0%) | 4 (28.6%) |  |
| **Education** |  |  |  |  | **0.432** |
| Secondary | 3 (2.97%) | 0 (0.00%) | 0 (0.00%) | 0 (0.00%) |  |
| College | 3 (2.97%) | 0 (0.00%) | 1 (12.5%) | 1 (7.14%) |  |
| University | 95 (94.1%) | 30 (100%) | 7 (87.5%) | 13 (92.9%) |  |
| **Time spent at home (hours/day)** |  |  |  |  | **0.364** |
| ≤8 | 14 (13.9%) | 1 (3.33%) | 1 (12.5%) | 2 (14.3%) |  |
| >8 | 87 (86.1%) | 29 (96.7%) | 7 (87.5%) | 12 (85.7%) |  |

p value not computed due to observed values.

COVID-19 mitigate initiatives and restrictive measures; we noticed that 86% of our participants desired financial assistance from the Government (Fig 3), and most of the participants agreed with the Government penalizing measures (Fig 4).

## Discussion

In the time where most countries worldwide have been coping with COVID-19 pandemic, Guinea is also struggling to do so. Although the physical effects of COVID-19 pandemic are increasingly well known, it is important to recognize its psychosocial impact as well [1, 12, 17, 18]. Responses to this pandemic hinge, on a great extent, to the capability of people to manage stress, worry and other psychosocial issues. To address COVID-19 pandemic, the Guinean Government has implemented unprecedented measures. Our study tried to identify the consequences of those measures on guinea population to inform policymakers.

Although the response rate was slightly low for IES-R (56%), our results showed that people had been differently stressed participants reported 19.6% (mild), 5.23% (moderate), and 9.15% (severe). These findings are similar to those found in a previous study [13]. The high rate of worry amongst participants could be related to the significant proportion of our study sample

**Table 4. Bivariate analysis Penn State Worry Questionnaire (PSWQ) and sociodemographic characteristics.**

|  | Low worry | Moderate worry | p-value |
|---|---|---|---|
|  | N = 154 | N = 32 |  |
| **Age** |  |  | **0.693** |
| 18–39 | 144 (93.5%) | 31 (96.9%) |  |
| 40–62 | 10 (6.49%) | 1 (3.12%) |  |
| **Gender** |  |  | **0.591** |
| Male | 102 (66.2%) | 19 (59.4%) |  |
| Female | 52 (33.8%) | 13 (40.6%) |  |
| **Marital Status** |  |  | **0.118** |
| Married | 59 (38.3%) | 7 (21.9%) |  |
| Single | 95 (61.7%) | 25 (78.1%) |  |
| **Residence** |  |  | **0.214** |
| Dixinn | 10 (6.49%) | 1 (3.12%) |  |
| Matam | 1 (0.65%) | 0 (0.00%) |  |
| Matoto | 40 (26.0%) | 15 (46.9%) |  |
| Ratoma | 82 (53.2%) | 12 (37.5%) |  |
| *Other | 21 (13.6%) | 4 (12.5%) |  |
| **Occupation** |  |  | **0.371** |
| Civil servant | 23 (14.9%) | 4 (12.5%) |  |
| Freelance | 33 (21.4%) | 5 (15.6%) |  |
| Housewife | 0 (0.00%) | 1 (3.12%) |  |
| Private sector employee | 27 (17.5%) | 7 (21.9%) |  |
| Unemployed | 71 (46.1%) | 15 (46.9%) |  |
| **Education** |  |  | **0.594** |
| Secondary | 2 (1.30%) | 1 (3.12%) |  |
| College | 5 (3.25%) | 1 (3.12%) |  |
| University | 147 (95.5%) | 30 (93.8%) |  |
| **Time spent at home (hours)/day** |  |  | **0.082** |
| ≤8 | 16 (10.4%) | 7 (21.9%) |  |
| >8 | 138 (89.6%) | 25 (78.1%) |  |

*Participants from outside Conakry capital city.

mainly composed of people from Ratoma municipality; this area was the most affected by COVID-19 at the time of the survey. Another relevant reason related to the occurrence of traumatic events is that nearly half of the participants were unemployed (Table 1) or had recently lost their job (Fig 2); these situations increase the level of distressful events since participants were staying at home for longer than usual (Table 1). Traumatic events of this study dovetail with the pre-pandemic situation. For instance, a study on Ebola survivors reported 15% of post-traumatic stress disorder revealing the outbreak effects [22]. Similarly, another finding revealed that bank employees stressed at different levels (very low stress = 50.47%, stress down = 44.43%, high stress = 5.09%) [23]. Participants mainly sought COVID-19 information from social media (Facebook), then come information from WHO and private web sites. The large proportion of young participants (94.5%) may explain the previous results.

Our study found no significant association between IES-R, worry scales, and sociodemographic variables. The low number of respondents and sampling frame that might have led to a selection bias may explain the latter finding. As expected, our results showed that most participants had negative impacts of COVID-19. This perception is linked to the fact that more

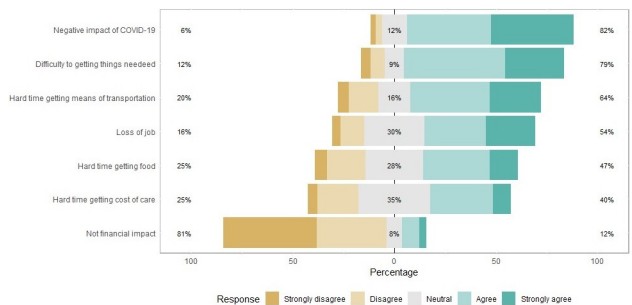

**Fig 2. COVID-19 economic impacts among Guinean population.**

than half of the participants had lost their job and had difficulties in meeting basic needs. Only a few of them (12%) enjoyed that time. It should be noted that even some employees were facing adverse impacts of COVID-19, due to the shutdown of other businesses that used to generate incomes. Participants wanted more support (like Government financial assistance) and requested more researches on COVID-19 by the Guinea Government. Our study sample was made up of relatively highly educated people (95%), and their opinion matched with those expectations (researches on COVID-19). For the records, the Government has made some efforts to mitigate COVID-19 effects, including making the public transport, domestic electricity, and running water free of charge for three (03) months (from March to May). Although people desired more from the Government, they instead upheld all the penalizing regulations regarding COVID-19. Given that most of our participants had a high level of education as we noticed in the study, they should easily agree with all these measures that prevent the spread of COVID-19.

The study is not representative of the overall population as its design did not take into account all social strata in Guinea. Taking part in an online survey is not an easy task for everyone, justifying some limitations in the conduct of the survey. Targeting people who are literate and have access to the internet induces a selection bias in the achievement of the study goals. In addition, even though people may have access to the internet, they are not familiar with an online survey. Otherwise, findings reported in this study might have been unlike. Another challenge was the difficulty to check for reliability of participants responses, even though the tools (IES-R and PSWQ) have shown good internal consistency in various studies [24–27]. Besides, social-psychological measurements of COVID-19 [21] have not yet been validated in Guinea. However, ascertainment of its internal validity showed good properties (S1 File).

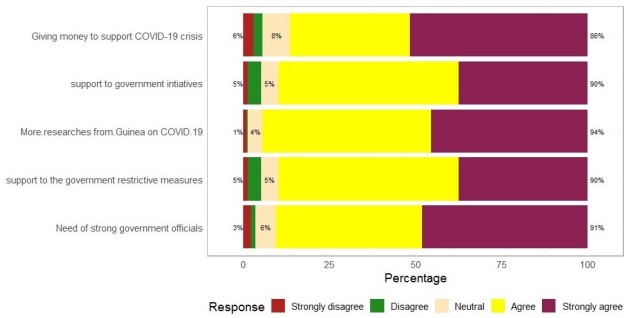

**Fig 3. Population opinions regarding government measures.**

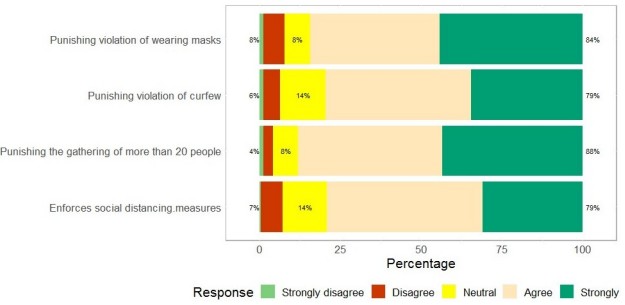

**Fig 4. Population perception of punishment against violators of preventive measures of COVID-19.**

Nonetheless, this study gives an overview of psychological dimensions in a selected population. The prevalence of traumatic events (stress and worry) indicates the need of further investigations, for psychological concerns would raise among the general population, health frontline workers, families and, patients affected by COVID-19 [8, 11, 28, 29]. Of note that a public protest took place around Conakry Capital city to denounce police brutality against violators of the Government preventive measures, such as masks wearing, the closure of worship places, and the curfew. This kind of behaviors highlights people's social and spiritual needs which go beyond the financial sustain. For a better public perception of COVID-19 and illiterate inclusion, a good alternative to the use internet would be the method of telephone interviews or community media (radio, TV).

## Conclusion

As the number of COVID-19 cases is increasing daily in Guinea, psychosocial concerns might also grow. People's mental state must be monitored to prevent unexpected events regarding the pandemic. Although there were few cases of traumatic events, our findings merit the attention of policymakers. Even though there were no significant statistical associations between sociodemographic variables and traumatic events (stress and worry), most participants were facing the adverse impacts of COVID-19; these effects could be potential predictors of psychosocial issues. Further investigations are therefore necessary to identify and disentangle specific psychosocial concerns in other Guinean socio-cultural contexts. In the meantime, information provide in this study can be used to set up coping strategies.

## Supporting information

**S1 File.**
(DOCX)

**S2 File.**
(ZIP)

## Acknowledgments

We are very grateful to the study participants for their availability; we also thank Matthew Brookoff –English teacher at YMCA New Americans Welcome Center (New York) and Andrée Prisca Ndjoug NDOUR, DVM, PhD, Assistante de recherche, Projet ZELS/ Brucelllose en Afrique de l'ouest et du centre Postdoc Afrique One ASPIRE for their valuable time to edit our manuscript.

## Author Contributions

**Conceptualization:** Almamy Amara Touré, Lansana Mady Camara, Aboubacar Sidiki Magassouba.

**Data curation:** Almamy Amara Touré, Aboubacar Sidiki Magassouba, Mohamed Sylla.

**Formal analysis:** Almamy Amara Touré, Gaspard Loua.

**Investigation:** Gnoume Camara, Alsény Yarie Camara, Diao Cissé.

**Methodology:** Almamy Amara Touré, Abdoulaye Doumbouya, Alpha Oumar Bereté.

**Project administration:** Lansana Mady Camara.

**Resources:** Almamy Amara Touré, Gnoume Camara.

**Software:** Mohamed Sylla.

**Supervision:** Lansana Mady Camara, Abdoulaye Doumbouya, Abdoul Habib Beavogui.

**Validation:** Almamy Amara Touré, Aboubacar Sidiki Magassouba.

**Writing – original draft:** Almamy Amara Touré, Abdoul Habib Beavogui.

**Writing – review & editing:** Almamy Amara Touré, Lansana Mady Camara, Aboubacar Sidiki Magassouba, Abdoulaye Doumbouya.

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
