## [Decision Letter · Decision Letter 0]

24 Nov 2020

PONE-D-20-19796

Psychosocial impacts of COVID-19 in the Guinean population. An Online cross-sectional survey.

PLOS ONE

Dear Dr. Almamy Amara Touré,

Thank you for submitting your manuscript to PLOS ONE. After careful consideration, we feel that it has merit but does not fully meet PLOS ONE’s publication criteria as it currently stands. Therefore, we invite you to submit a revised version of the manuscript that addresses the points raised during the review process.

We look forward to receiving your revised manuscript.

Kind regards,

Akihiro Nishi, M.D., Dr.P.H.

Academic Editor

PLOS ONE

Journal Requirements:

2. Please include additional information regarding the survey or questionnaire used in the study and ensure that you have provided sufficient details that others could replicate the analyses. For instance, if you developed a questionnaire as part of this study and it is not under a copyright more restrictive than CC-BY, please include a copy, in both the original language and English, as Supporting Information. Moreover, please include more details on how the questionnaire was pre-tested, and whether it was validated; and please provide more information on how variables were categorised.

3.We note that you have indicated that data from this study are available upon request. PLOS only allows data to be available upon request if there are legal or ethical restrictions on sharing data publicly. For information on unacceptable data access restrictions, please see http://journals.plos.org/plosone/s/data-availability#loc-unacceptable-data-access-restrictions.

4. Please ensure that you refer to Figure 1 in your text as, if accepted, production will need this reference to link the reader to the figure.

Additional Editor Comments (if provided):

Please follow the two reviewers' comments and submit the revised one.

Reviewers' comments:

Reviewer's Responses to Questions

**Comments to the Author**

1. Is the manuscript technically sound, and do the data support the conclusions?

Reviewer #1: Yes

Reviewer #2: Partly

2. Has the statistical analysis been performed appropriately and rigorously? 

Reviewer #1: No

Reviewer #2: Yes

3. Have the authors made all data underlying the findings in their manuscript fully available?

Reviewer #1: Yes

Reviewer #2: Yes

4. Is the manuscript presented in an intelligible fashion and written in standard English?

Reviewer #1: No

Reviewer #2: Yes

5. Review Comments to the Author

Reviewer #1: This paper explores the psychological effects of COVID-19 on the population of Guinea. The topic is interesting and worth of investigation. However, parts of the analysis can be improved. In addition, the presentation and typography of the paper could use further refinement. Some suggestions to improve the paper are as follows:

Major

- I would suggest a comparison of the results presented in the paper with pre-pandemic data on stress levels in the Guinea population to see how/if stress levels have risen following the pandemic.

- I suggest enlisting the help of a scientific technical writer who is familiar with the field

Minor

- In Figure 4, I would suggest using a more noticeable color for "Strongly Agree" as it is difficult to see the respective bar upon first glance

Reviewer #2: This is a cross-sectional study of the psychosocial impact of COVID-19 in the Guinean population. It recruited via social media and had a study population of 275 people. The response for the psychological measures were lower at 56% and two thirds (should give the n and % for both questionnaires rather than 2/3).

My main concerns with this study are sample size and whether the study population is representative of the Guinean population or are significantly biased.

An internet search gives estimates of average number of Facebook friends worldwide as 338 with a median of 200 (brandwatch.com). I would worry that with a sample size of 275 this could be a particularly restricted social group. How were internet users identified to receive the survey link - was this based on targeted demographics via advertisement or via peers/social contacts of the researchers?

It would be good to have an idea of how typical this sample is in terms of demographics compared with the general population of Guinea. There is a high level of university education in the unemployed group and I wonder how typical that would be in general or if this is a particularly vulnerable group in a pandemic.

The manuscript is mostly well written although there are some problems with tenses and some typographical errors eg

- abstract: methods: 'we did an online cross-sectional survey among people' - could change to "we carried out an online cross-sectional study amongst internet users in Guinea"

- introduction - take out 'the' before global health, distanciation is referred to more commonly as "social distancing", capital city was (not were) isolated, first time to face more usually "first time facing", 2interested in to" could b e "aimed to"

6. PLOS authors have the option to publish the peer review history of their article (what does this mean?). If published, this will include your full peer review and any attached files.

Reviewer #1: No

Reviewer #2: No

---

## [Author Response · Author response to Decision Letter 0]

8 Dec 2020

2020/12/07

Dear Akihiro Nishi, M.D, Dr P.H.

Academic Editor

PLOS ONE

Subject: Submission of revised paper PONE-D-20-19796 (manuscript ID number that the journal assigned).

Dear Dr Nishi,

First of all, we would like to thank you for your email of November 24th, 2020, and the opportunity to resubmit the revised version of our manuscript. We appreciate the time and effort that you and the reviewers dedicated to providing feedback on our manuscript and are grateful for your insightful and valuable comments on our paper.

We have incorporated the suggestions made by the reviewers. Our responses are given in a point-by-point manner in italic font. Changes to the manuscript are highlighted in yellow font. In this revised version of the manuscript, we did our best to address all comments made by the Reviewers. 

We hope that the Reviewers and the Editors will be satisfied with our responses to their "comments".

Sincerely yours,

Almamy Amara TOURE

Head of monitoring/Evaluation unit

National Centre for Training and Research in Rural Health of Maferinyah, Forecariah, Guinea.

On behalf of the authors

Reviewer #1: This paper explores the psychological effects of COVID-19 on the population of Guinea. The topic is interesting and worth of investigation. However, parts of the analysis can be improved. In addition, the presentation and typography of the paper could use further refinement. Some suggestions to improve the paper are as follows:

Major

- I would suggest a comparison of the results presented in the paper with pre-pandemic data on stress levels in the Guinea population to see how/if stress levels have risen following the pandemic.

- I suggest enlisting the help of a scientific technical writer who is familiar with the field

Response

Thank you for having raised that question; we believe it was worth to be mentioned. For the records, our country (Guinea) has been facing a couple of health crises such as cholera, meningitis, Ebola, and now COVID-19. Before undertaking this study, we did an in-depth review of the literature to look at what has been done. Unfortunately, we found no specific information about psychosocial impact in the Guinea general population during the pre epidemic or pandemic period in Guinea. However, to reply to your thoughtful remark, we inserted in the discussion section as follow: “Traumatic events of this study dovetail with the pre-pandemic situation. For instance, a study on Ebola survivors reported 15% of post-traumatic stress disorder revealing the outbreak effects[]. Similarly,another finding revealed that bank employees stressed at different levels (very low stress=50.47%,stress down=44.43% ,high stress=5.09%).” (page 12, from line 8 to 11).

Minor

- In Figure 4, I would suggest using a more noticeable color for "Strongly Agree" as it is difficult to see the respective bar upon first glance.

Response 

Thank you for this observation

We changed colour for the label "Strongly Agree", you can notice in the manuscript (page number 12).

Reviewer #2: This is a cross-sectional study of the psychosocial impact of COVID-19 in the Guinean population. It recruited via social media and had a study population of 275 people. The response for the psychological measures were lower at 56% and two thirds (should give the n and % for both questionnaires rather than 2/3).

Response

Thank you for this remark

We gave the n and % in the appropriate place.

My main concerns with this study are sample size and whether the study population is representative of the Guinean population or are significantly biased.

An internet search gives estimates of average number of Facebook friends worldwide as 338 with a median of 200 (brandwatch.com). I would worry that with a sample size of 275 this could be a particularly restricted social group. How were internet users identified to receive the survey link - was this based on targeted demographics via advertisement or via peers/social contacts of the researchers?

It would be good to have an idea of how typical this sample is in terms of demographics compared with the general population of Guinea. There is a high level of university education in the unemployed group and I wonder how typical that would be in general or if this is a particularly vulnerable group in a pandemic.

Response

Thank you for your comments; this is an interesting observation.

As we have mentioned in the manuscript, participants came from everywhere as shows in this statement "A free online e-survey ( https://esurv.org ) platform was used to send questionnaires to the internet users, they were encouraged to pass it to each other through social media (Facebook, WhatsApp, messenger) by using snowball sampling". Each author sent a questionnaire through groups of social media organize not only the researcher, and then they were encouraged to pass it to each other. Of note that we recoded the variable profession so that it can be more readable; for we noticed more than 100 categories socio-professionals accounting for all potential users of the Internet in Guinea. Another reason to consider our sample not representative of our entire population is based on the fact that not everyone uses the Internet in Guinea. Even among the literate people, Internet use is minimal. Besides, the spreading of the internet is relatively new (less than 10 years) in Guinea. Dear reviewer to address your concerns, we added more explanations in the description of the study population in the manuscript as follow:" The study population were diverse and came from all the sectors of activities in Guinea, but essentially focused on those who were active in using the internet, for the period of the study was short" (page number 3 from line 4 to 6 ), we also precise the social group that our survey focused on in the discussion section “ Targeting people who are literate and have access to the internet introduces a selection bias in the achievement of the study” (page number13, second paragraph, from line 3 to 4).

The manuscript is mostly well written although there are some problems with tenses and some typographical errors eg

- abstract: methods: 'we did an online cross-sectional survey among people' - could change to "we carried out an online cross-sectional study amongst internet users in Guinea"

- introduction - take out 'the' before global health, distanciation is referred to more commonly as "social distancing", capital city was (not were) isolated, first time to face more usually "first time facing", 2interested in to" could be "aimed to"

Response

Thank you for this valuable input.

We corrected all your observations;

-Abstract : methods: we replaced "did" by “carried out” the first line of the section.

-Introduction: “the” taken out (page 2 from line 5 to 6) , “distanciation changed” in “social distancing” (introduction, page number 2, second paragraph, line 4), capital city was inserted (introduction, page number 2, second paragraph, line 5), "first time facing" inserted instead of “first time to face” (introduction, page number 2, third paragraph, line 5), "aimed to" inserted instead of “interested in” (introduction, page number 2, last paragraph).

---

## [Decision Letter · Decision Letter 1]

7 Jan 2021

Psychosocial impacts of COVID-19 in the Guinean population. An online cross-sectional survey.

PONE-D-20-19796R1

Dear Dr. Almamy Amara Touré,

We’re pleased to inform you that your manuscript has been judged scientifically suitable for publication and will be formally accepted for publication once it meets all outstanding technical requirements.

Kind regards,

Akihiro Nishi, M.D., Dr.P.H.

Academic Editor

PLOS ONE

Additional Editor Comments (optional):

I am please to recommend an accept and an immediate publication.

Reviewers' comments:

Reviewer's Responses to Questions

**Comments to the Author**

1. If the authors have adequately addressed your comments raised in a previous round of review and you feel that this manuscript is now acceptable for publication, you may indicate that here to bypass the “Comments to the Author” section, enter your conflict of interest statement in the “Confidential to Editor” section, and submit your "Accept" recommendation.

Reviewer #1: All comments have been addressed

Reviewer #2: All comments have been addressed

2. Is the manuscript technically sound, and do the data support the conclusions?

Reviewer #1: (No Response)

Reviewer #2: Yes

3. Has the statistical analysis been performed appropriately and rigorously? 

Reviewer #1: (No Response)

Reviewer #2: Yes

4. Have the authors made all data underlying the findings in their manuscript fully available?

Reviewer #1: (No Response)

Reviewer #2: Yes

5. Is the manuscript presented in an intelligible fashion and written in standard English?

Reviewer #1: (No Response)

Reviewer #2: Yes

6. Review Comments to the Author

Reviewer #1: (No Response)

Reviewer #2: This is an important and timely topic which adds to the international literature on the impact of COVID-19 in a country that has experienced the trauma of other significant outbreaks of illness.

The authors have stated the limitations around sampling more clearly.

The standard of written English in the paper has been improved.

7. PLOS authors have the option to publish the peer review history of their article (what does this mean?). If published, this will include your full peer review and any attached files.

Reviewer #1: No

Reviewer #2: No

---

## [Editor Report · Acceptance letter]

22 Jan 2021

PONE-D-20-19796R1 

Psychosocial impacts of COVID-19 in the Guinean population. An online cross-sectional survey. 

Dear Dr. Touré:

I'm pleased to inform you that your manuscript has been deemed suitable for publication in PLOS ONE. Congratulations! Your manuscript is now with our production department. 

Kind regards, 

on behalf of

Dr. Akihiro Nishi 

Academic Editor

PLOS ONE